behaviour/ecology

animal tracking, bird migration, Brownian bridge movement models, habitat displacement, soaring birds, wind farms

**Author for correspondence:**
Carlos David Santos
e-mail: cdsantos@ufpa.br

# Black kites of different age and sex show similar avoidance responses to wind turbines during migration

Carlos David Santos[1,2], Rafael Ferraz[1],
Antonio-Román Muñoz[3], Alejandro Onrubia[4]
and Martin Wikelski[2,5]

[1]Núcleo de Teoria e Pesquisa do Comportamento, Universidade Federal do Pará, Rua Augusto Correa 01, Guamá, 66075-110 Belém, Brazil
[2]Department of Migration, Max Planck Institute for Animal Behavior, Am Obstberg 1, 78315 Radolfzell, Germany
[3]Biogeography, Diversity and Conservation Research Team, Departamento de Biología Animal, Facultad de Ciencias, Universidad de Málaga, Spain
[4]Fundación Migres (CIMA), Ctra. N-340, Km.85, Tarifa, 11380 Cádiz, Spain
[5]Department of Biology, University of Konstanz, Universitätsstr. 10, 78464 Konstanz, Germany

CDS, 0000-0001-5693-9795

Populations of soaring birds are often impacted by wind-power generation. Sex and age bias in turbine collisions can exacerbate these impacts through demographic changes that can lead to population decline or collapse. While several studies have reported sex and age differences in the number of soaring birds killed by turbines, it remains unclear if they result from different abundances or group-specific turbine avoidance behaviours, the latter having severer consequences. We investigated sex and age effects on turbine avoidance behaviour of black kites (*Milvus migrans*) during migration near the Strait of Gibraltar. We tracked the movements of 135 individuals with GPS data loggers in an area with high density of turbines and then modelled the effect of proximity of turbines on bird utilization distribution (UD). Both sexes and age classes showed similar patterns of displacement, with reduced UD values in the proximity of turbines and a clear peak at 700–850 m away, probably marking the distance at which most birds turn direction to avoid approaching the turbines further. The consistency of these patterns indicates that displacement range can be used as an accurate proxy for collision risk and habitat loss, and should be incorporated in environmental impact assessment studies.

# 1. Introduction

Countries around the world are taking action against climate change by shifting to renewable energy sources. This prompted a rapid development of wind-power industry over the past two decades, with the occupation of large natural areas by wind-power plants [1]. Conflicts between wind-power generation and wildlife are well documented, with birds and bats being the most impacted groups through direct mortality by collision with turbines and the displacement from areas vital for their survival [2]. Terrestrial soaring birds, including most raptors, storks, pelicans and other broad-winged large birds, raise particular concerns as their flight is favoured by landscape characteristics similar to those targeted by wind-power developers, i.e. mountain ridges and slopes at regions of frequent winds [3,4]. In addition, soaring birds have low flight manoeuverability, increasing their chances of collision with turbines, and low fecundity, limiting their capacity to out-balance additional mortality [2].

Sex- and age-skewed mortality gradually changes the demographic structure of animal populations, which can lead to accelerated declines or even population collapses [5,6]. Thus, understanding the full impact of wind-power generation on species that collide with turbines critically depends on identifying potential sex- and age-related biases on collision risk. Several studies have reported sex and age differences in the numbers of soaring birds killed by wind turbines [e.g. 7–11]. However, it remains unclear whether these differences reflect vulnerability of a particular sex or age class to collisions or simply unequal relative abundances. Nevertheless, sex- and age-related differences in behaviour and ecological requirements of soaring birds have the potential to influence their collision risk with wind turbines. During the breeding season, the time allocated to incubation decreases the risk of collision of females [11,12]. The engagement of adult males with territory defence increases frequency of social interactions during which they might have limited awareness of turbine collision risk [13]. Subadults and floaters may occupy vacant territories close to turbines increasing their chances of colliding with turbines [10,13]. The poorer flight ability of juveniles [14,15] and limited perception of danger [16] may place them at a higher risk of collision.

We specifically investigated how sex and age may influence turbine avoidance behaviour using black kites (*Milvus migrans*) as a model species of soaring birds. Turbine avoidance was quantified as the displacement of bird activity in the proximity of turbines, which directly influences collision risk and habitat loss [17,18]. Displacement effect was evaluated from high-frequency GPS tracking data of 135 individual black kites relatively balanced among sexes (61 males and 72 females) and age classes (77 adults or 58 juveniles). Birds were tracked during the post-breeding migration before they crossed the Strait of Gibraltar, when they use an area with high turbine density. We expected males and females to show similar avoidance of turbines because no prior evidence supports sex differences in space use or other relevant behaviour for this species or other soaring birds during migration. However, we expected juveniles to show reduced avoidance of turbines given their general lack of experience and the high juvenile fatality rates reported for other raptor species in this area during migration [7,9].

# 2. Material and methods

## 2.1. Data collection

We collected GPS-tracking data from 135 black kites moving in the region of Tarifa (Southern Spain) during their post-breeding migration. Birds were caught with a walk-in trap ($7 \times 7 \times 3.5$ m) during periods of high-speed crosswinds at the Strait of Gibraltar, which restrict their passage to Africa [15,19]. Such conditions can last for periods up to a week [20] forcing the birds to roam in an area with high density of wind turbines (figure 1). Birds were captured in 2012 and 2013 between July and September. In each capture, we tagged similar numbers of juveniles and adults in order to produce an age-balanced sample (77 adults and 58 juveniles). Birds were aged from plumage patterns (following [21]). Molecular sexing conducted from breast feather samples (following [22]) showed that both sexes were also similarly represented in our sample (61 males and 72 females). Birds were tagged with GPS-GSM data loggers (42 g, TM-202/R9C5 module; Movetech Telemetry, UK, http://movetech-telemetry.com) attached as backpacks using Teflon ribbon harnesses. The dataloggers recorded GPS fixes every minute from 9.30 to 18.30, and every 20 min during the early and late hours of the day (7.30 to 9.30 and 18.30 to 20.30) when birds were less likely to fly. Loggers recorded data with a higher resolution (GPS fixes every 10 s with 20 s bursts at 1 Hz every 3 min) when birds got close to the edge of the Strait of Gibraltar. Logged data was transmitted to an Internet server via GPRS every 2 h.

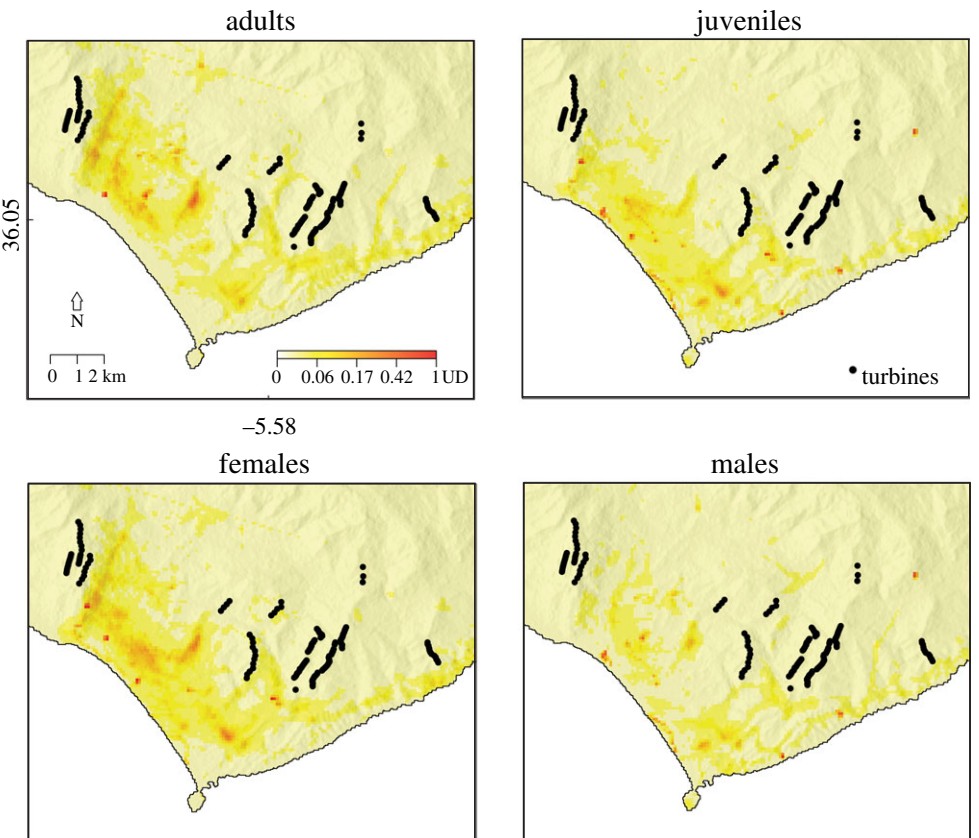

**Figure 1.** Utilization distribution (UD) of black kites in the study area (Tarifa, Spain) discriminated by sex or age. UD values are shown in a colour gradient, with darker colours reflecting higher UD. All plots follow the UD scale presented in top left plot. UD values resulted from dynamic Brownian bridge movement models (dBBMMs) built with GPS tracking data of 135 birds caught during the post-breeding migration in 2012 and 2013. UD resolution is 100 × 100 m. Hill shading was added to image background to show interaction between bird movement and topography.

## 2.2. Data analysis

The effect of turbine proximity on bird utilization distribution (UD) was modelled with generalized additive mixed models (GAMMs), as this relationship was shown before to be nonlinear [17]. Bird UD was produced from the tracking data using dynamic Brownian bridge movement models (dBBMMs, [23]). This method accounts with the time between locations in UD estimation, which is major improvement in relation to kernel-based methods, commonly employed for space use inference [23,24]. The dataset used for dBBMMs included only GPS fixes of birds in flight (with speed greater than 1 m s$^{-1}$) and recorded in a target area with high bird movement and turbine density (electronic supplementary material, figure S1). We also restricted the data to those collected during easterly winds (direction 70 to 130°), representing *ca* 90% of all data recorded. Other wind conditions allowed birds to quickly cross the Strait of Gibraltar [15,19], limiting the potential of the tracking data recorded for the purposes of this study. dBBMMs were produced for each individual bird in each day in a 100 × 100 m grid, and these models were then summed up to produce a general UD. The dBBMMs were built with the function brownian.bridge.dyn of the R package move [25], using a window size of 15 locations, a margin of five locations and a location error of 20 m. Besides proximity to turbines, the GAMMs included orographic and thermal uplift as predictors of UD since these variables are critical for the movement of soaring birds [26]. Orographic and thermal uplift were mapped for the study area following the methods described in Santos *et al*. [26]. The estimation of orographic uplift velocity uses terrain aspect and slope, which were extracted from a 30 m resolution digital elevation model (available at https://lpdaac.usgs.gov), and wind direction and speed, which were obtained from local weather stations. The methods of Santos *et al*. [26] estimate thermal uplift velocity from land surface temperature, which is retrieved from Landsat imagery. For that purpose, we used a Landsat 8 OLI/TIRS image acquired on 17 July 2013 (available at https://earthexplorer. usgs.gov), matching the period of data collection. Orographic and thermal uplift velocities were

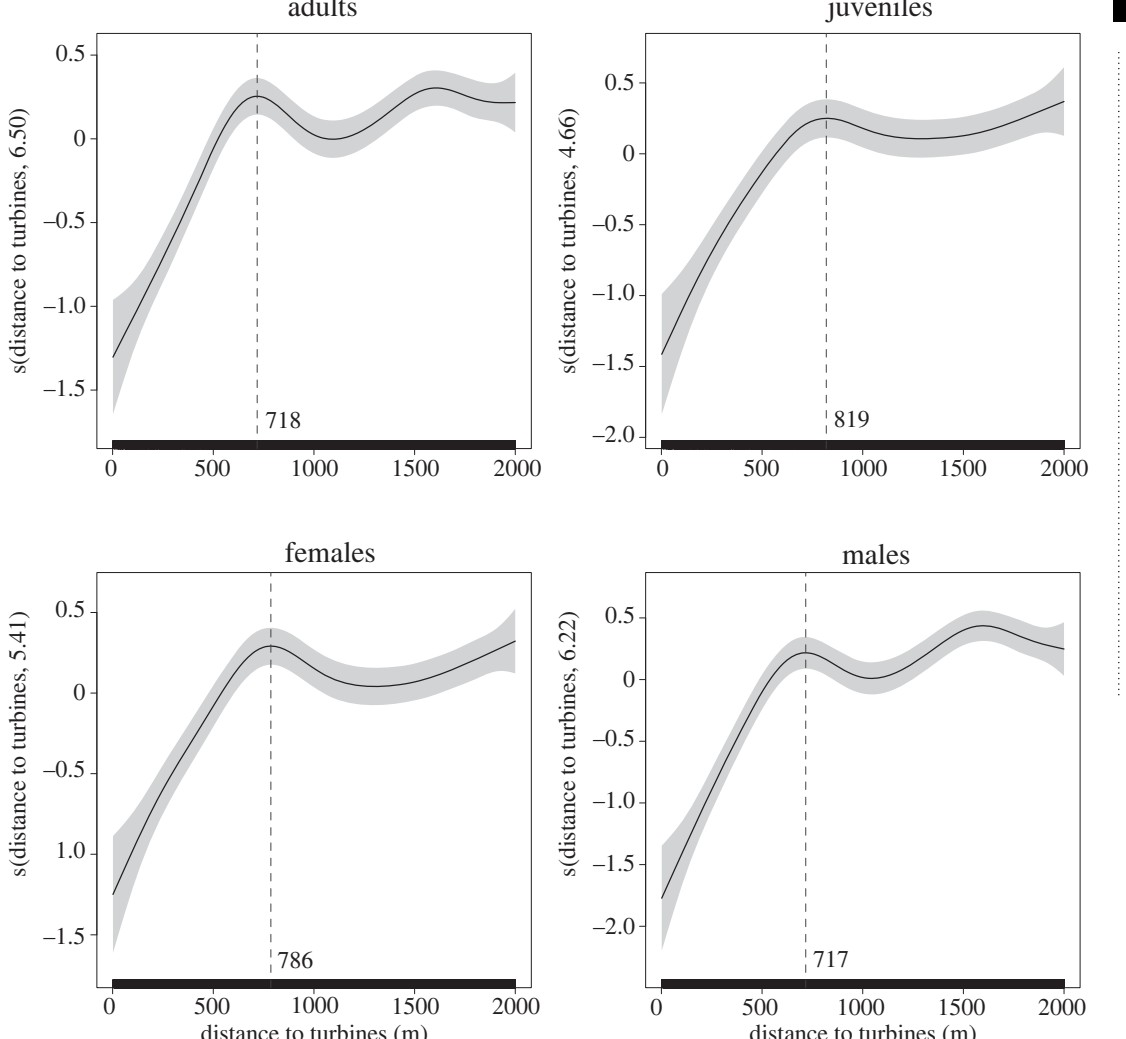

**Figure 2.** GAMM partial effects of turbine proximity on utilization distribution (UD) of black kites. Different models were built for each sex and age class. All four models included orographic and thermal uplift velocities as predictors, and accounted for spatial autocorrelation. Shaded areas represent 95% confidence intervals.

included as linear predictors in the GAMMs, as they tend to show a linear relationship with bird UD [17]. GAMMs were fitted with the function gamm of the R package mgcv [27], assuming Tweedie error distribution (log link and power variance = 1.6) and Gaussian spatial correlation structure, in order to account for spatial autocorrelation. Fitting assumptions were checked from residual plots using the function gam.check of the mgcv package. Correlations between model predictors were lower than 0.2. GAMMs were produced with data of grid cells at distances up to 2 km from wind turbines to avoid confounding effects of factors possibly acting at larger scales. A few extreme UD values were removed prior to analysis to prevent their overinfluence in our results (three for the juvenile model, two for the female model and one for the male model). The modelling results including these data are presented in electronic supplementary material, figure S2.

## 3. Results

In general, bird movements tended to concentrate in a belt of *ca* 5 km inland from the shoreline, and particularly in a valley on the east side of the study area (figure 1). Juveniles seem to move particularly close to the shoreline. In all four groups, there was a visible drop in movement density in the areas nearby wind turbines (figure 1).

There was a clear nonlinear effect of turbine distance on bird UD for all four groups compared, with UD values dropping abruptly in the first few hundred meters of the wind turbines (figure 2). The relationship had

**Table 1.** Summary of GAMMs relating UD of black kites with turbine proximity and orographic and thermal uplift velocities. Different models were built for each sex and age class. Models were fitted with Tweedie error distribution (log link and power variance = 1.6) and Gaussian spatial correlation structure, in order to account for spatial autocorrelation. s.e., Standard error; $t$, T statistics; edf, estimated degrees of freedom; $F$, F statistics.

| model | parameter | estimate | s.e. | $t$ | edf | $F$ | $p$-value | $R^2$adj. |
|---|---|---|---|---|---|---|---|---|
| adults | intercept | −12.2 | 0.40 | −30.3 | | | $<2 \times 10^{-16}$ | 0.15 |
| | s(distance to turbines) | | | | 6.50 | 24.8 | $<2 \times 10^{-16}$ | |
| | orographic uplift | 0.3 | 0.02 | 14.3 | | | $<2 \times 10^{-16}$ | |
| | thermal uplift | 3.4 | 0.21 | 16.1 | | | $<2 \times 10^{-16}$ | |
| juveniles | intercept | −10.8 | 0.58 | −18.5 | | | $<2 \times 10^{-16}$ | 0.05 |
| | s(distance to turbines) | | | | 4.66 | 16.8 | $2.8 \times 10^{-15}$ | |
| | orographic uplift | 0.2 | 0.04 | 4.9 | | | $1.2 \times 10^{-16}$ | |
| | thermal uplift | 2.6 | 0.31 | 8.4 | | | $<2 \times 10^{-16}$ | |
| females | intercept | −12.8 | 0.47 | −27.0 | | | $<2 \times 10^{-16}$ | 0.14 |
| | s(distance to turbines) | | | | 5.41 | 18.1 | $<2 \times 10^{-16}$ | |
| | orographic uplift | 0.3 | 0.03 | 10.0 | | | $<2 \times 10^{-16}$ | |
| | thermal uplift | 3.8 | 0.25 | 15.1 | | | $<2 \times 10^{-16}$ | |
| males | intercept | −9.8 | 0.47 | −21.0 | | | $<2 \times 10^{-16}$ | 0.07 |
| | s(distance to turbines) | | | | 6.22 | 28.9 | $<2 \times 10^{-16}$ | |
| | orographic uplift | 0.2 | 0.03 | 8.6 | | | $<2 \times 10^{-16}$ | |
| | thermal uplift | 2.0 | 0.25 | 8.1 | | | $6 \times 10^{-16}$ | |

a distinct peak that was relatively similar in all groups (700–850 m) after which it tended to stabilize (figure 2). A second peak was present around 1500 m from the wind turbines for adults and males, although this might be an artefact that results from excessive degrees of freedom in the smoother (figure 2). These patterns remained the same when testing groups of independent individuals (adult females, adult males, juvenile females and juvenile males, electronic supplementary material, figures S3 and S4).

The effects of orographic and thermal uplift velocities on bird UD were significant and with increasing trends in all four models (table 1).

## 4. Discussion

We found no relevant differences in the patterns of turbine avoidance between sexes or age classes of migrating black kites. The effect of turbine proximity on bird UDs had a consistent pattern among the groups compared, with a gradual increase of UD up to 700–850 m from the turbine locations (hereafter displacement range) and after a slight decrease UD values tended to stabilize (figure 2). The consistency of this pattern among groups of birds that travelled in different areas and had interactions with different turbines suggests it results from a systematic avoidance behaviour of individuals. Birds probably kept a similar safe distance from turbines and the peak observed in UD seems to mark the distance at which most birds turned direction to avoid approaching the turbines further. This phenomenon was observed before in raptors tracked by radar (see fig. 4 of [28]). Results from other studies suggest that this avoidance pattern is common among soaring birds, although the displacement range varies between species and across the annual cycle [29–31].

Confirming our earlier expectations, female and male black kites showed similar patterns of turbine avoidance. However, it is interesting to note that the displacement range was slightly higher in females than males (786 and 717 m, respectively). Whether these differences have biological relevance is difficult to ascertain, but females are expected to be less manoeuverable than males due to their larger body size [2,11], which may keep them at a larger distance from turbines. The similarity in the avoidance behaviour of juvenile and adult black kites was rather surprising. Previous studies reporting of high juvenile collision rates in other raptor species in this exact area and season suggested that juveniles could

approach wind turbines closer, either because of being naive or having lower manoeuverability than adults [7,9]. However, the small differences that we found in the displacement range between juveniles and adults follow an opposite trend, with adults getting closer to turbines than juveniles.

Our results add to the previous knowledge that turbine avoidance behaviour of soaring birds during migration may not be affected by sex or age, contrary to observations in other studies during the breeding season [10,11,13]. Obviously, further studies are needed to ascertain if our results apply to other soaring bird species and spatio-temporal contexts. Importantly, we showed a consistent pattern of turbine displacement among the different groups of birds tested. Similar patterns with varying displacement ranges were also reported in earlier studies [29–31]. Displacement ranges should be further investigated and incorporated in environmental impact assessment studies, as they can help predicting collision risk and habitat loss [17,18]. Our results also elucidate that before–after, control–impact (BACI) studies comparing areas wider than species-specific displacement ranges may not be able to detect the effects of wind turbines on bird densities. The accurate determination of the displacement ranges in this study was possible due to high-resolution data provided by GPS telemetry and the use of a modelling approach with the power to discriminate complex ecological relationships. We recommend similar approaches in future studies to provide the necessary empirical evidence for a complete understanding of the consequences of turbine avoidance by soaring birds.

Ethics. Bird trapping and the GPS tagging were approved by the Consejería de Medio Ambiente of the Junta de Andalucía through license to Alejandro Onrubia.

Data accessibility. Data available from Movebank Data Repository: https://doi.org/10.5441/001/1.23n2m412.

Authors' contributions. C.D.S. and M.W. designed the study; C.D.S., A.-R.M. and A.O. collected the data; C.D.S. and R.F. analysed the data; C.D.S. wrote the manuscript. All authors discussed the results and commented on the manuscript.

Competing interests. We declare we have no competing interests.

Funding. This research was funded by the Portuguese Foundation for Science and Technology (grant no. SFRH/BPD/64786/2009), the Deutsche Forschungsgemeinschaft (Germany's Excellence Strategy—EXC 2117–422037984), Programa Operativo FEDER Andalucía (grant no. FEDERJA-276) and the Max Planck institute für Verhaltensbiologie.

Acknowledgements. We thank Fundacion Migres staff for the logistic support during the fieldwork and the volunteers who helped with the bird captures. We also thank João Paulo Silva for help with the tracking methods and Hariprasath Ramesh for comments on the manuscript.

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
