## [Reviewer comments · Royal Society Open Science]

Review History

RSOS-201933.R0 (Original submission)

Review form: Reviewer 1

Is the manuscript scientifically sound in its present form?

Yes

Are the interpretations and conclusions justified by the results?

Yes

Is the language acceptable?

Yes

Do you have any ethical concerns with this paper?

No

Have you any concerns about statistical analyses in this paper?

No

Recommendation?

Accept with minor revision (please list in comments)

Comments to the Author(s)

Dear authors and editor,

The authors have made some minor changes to the manuscript.

However, they have responded in detail to my original comments in their response letter. My only recommendation would be to include some of this material in the supplementary information. In addition it might also be useful to include some of the analysis code in the supplementary information. Seeing that this uses existing R packages, it should not be too onerous and it will greatly facilitate replication, alongside the published data.

Other than that this work is technically sound and contributes to an important discussion in research with immediate real-world importance.

Kind regards.

Review form: Reviewer 2

Is the manuscript scientifically sound in its present form?

Yes

Are the interpretations and conclusions justified by the results?

Yes

Is the language acceptable?

Yes

Do you have any ethical concerns with this paper?

No

Have you any concerns about statistical analyses in this paper?

No

Recommendation?

Accept as is

Comments to the Author(s)

The authors did a great job in refuting the reviewers comments and amended the manuscript satisfactorily. I have no further comments on their revised manuscript.

Review form: Reviewer 3 (Robert Simmons)

Is the manuscript scientifically sound in its present form?

Yes

Are the interpretations and conclusions justified by the results?

Yes

Is the language acceptable?

Yes

Do you have any ethical concerns with this paper?

No

Have you any concerns about statistical analyses in this paper?

No

Recommendation?

Accept with minor revision (please list in comments)

Comments to the Author(s)

This revised version of the original ms is much crisper and succinct than the first version. I liked your response to my criticism that it may not be of general interest to the scientific community - based on the impressive citation record you give for a similar paper on this topic.

This has become a good, short, succinct contribution to the literature on this topic.

There are a few tenses that need attention (e.g. you new legend to the Figure 1 on the Utilization Distributions).

I cannot comment on the statistical component of the paper. Nice work.

Decision letter (RSOS-201933.R0)

This year has been very difficult for everyone, and we want to take the opportunity to thank you for your continued support in 2020.

The Royal Society Open Science editorial office will be closed from the evening of Friday 18 December 2020 until Monday 4 January 2021. We will not be responding during this time. If you have received a deadline within this time period, please contact us as soon as possible to allow us to extend the deadline. If you receive any automated messages during this time asking you to meet a deadline, we offer apologies and invite you to respond after the festive period or during normal working hours.

With our best for a peaceful festive period and New Year, and we look forward to working with you in 2021.

Dear Dr Santos

On behalf of the Editors, we are pleased to inform you that your Manuscript RSOS-201933 "Black kites of different age and sex show similar avoidance responses to wind turbines during migration" has been accepted for publication in Royal Society Open Science subject to minor revision in accordance with the referees' reports. Please find the referees' comments along with any feedback from the Editors below my signature.

Please submit your revised manuscript and required files (see below) within 1 month. Note: the ScholarOne system will 'lock' if submission of the revision is attempted 7 or more days after the deadline. If you do not think you will be able to meet this deadline please contact the editorial office immediately.

on behalf of Dr Sean Rands (Associate Editor) and Pete Smith (Subject Editor)
openscience@royalsociety.org

Associate Editor Comments to Author (Dr Sean Rands):

Your manuscript was re-reviewed by the three original reviewers for the Biology Letters submission, and their very positive comments should be attached below. Based on their comments plus my own non-specialist reading of the manuscript, I think this is now good for publication. I will ask for one small addition however: I agree with reviewer #1 that you should be providing at least some example analysis code to accompany the data you have made available, which should make your work more open (and which ties in with journal policy concerning code availability and statistical tools).

Reviewer comments to Author:

Reviewer: 1
Comments to the Author(s)

Dear authors and editor,

The authors have made some minor changes to the manuscript. However, they have responded in detail to my original comments in their response letter. My only recommendation would be to include some of this material in the supplementary information. In addition it might also be useful to include some of the analysis code in the supplementary information. Seeing that this uses existing R packages, it should not be too onerous and it will greatly facilitate replication, alongside the published data. Other than that this work is technically sound and contributes to an important discussion in research with immediate real-world importance.

Kind regards.

Reviewer: 2

Comments to the Author(s)

The authors did a great job in refuting the reviewers comments and amended the manuscript satisfactorily. I have no further comments on their revised manuscript.

Reviewer: 3

Comments to the Author(s)

This revised version of the original ms is much crisper and succinct than the first version. I liked your response to my criticism that it may not be of general interest to the scientific community - based on the impressive citation record you give for a similar paper on this topic.

This has become a good, short, succinct contribution to the literature on this topic.

There are a few tenses that need attention (e.g. you new legend to the Figure 1 on the Utilization Distributions).

I cannot comment on the statistical component of the paper. Nice work.

===PREPARING YOUR MANUSCRIPT===

===PREPARING YOUR REVISION IN SCHOLARONE===

Author's Response to Decision Letter for (RSOS-201933.R0)

See Appendix A.

Decision letter (RSOS-201933.R1)

Dear Dr Santos,

It is a pleasure to accept your manuscript entitled "Black kites of different age and sex show similar avoidance responses to wind turbines during migration" in its current form for publication in Royal Society Open Science.

on behalf of Dr Sean Rands (Associate Editor) and Pete Smith (Subject Editor)
openscience@royalsociety.org

Appendix A

Dear editor,

As encouraged by your response, we hereby submit a revised version of our manuscript RSOS-201933. We followed the advice of the associate Editor and Reviewer #1, and included R scripts for the main steps of data analysis in supplementary information. We also included two additional figures in supplementary information following the suggestion of Reviewer #1. Figure S1 illustrates how the dataset used for modelling fits with the full dataset available in Movebank. Figure S2 replicates figure 2 without the removal of UD outliers. Minor changes were also made in the manuscript: references to supplementary figures; reference to Movebank dataset; grammar corrections in legend of figure 1. We also provide a manuscript version with tracked changes. We hope these changes meet your expectations, but further suggestions to make changes to the manuscript will be welcome, should you deem them necessary.

Sincerely,

Carlos David Santos